# QUANTIFYING EXPOSURE BIAS
# FOR OPEN-ENDED LANGUAGE GENERATION

## ABSTRACT

The exposure bias problem refers to the incrementally distorted generation induced by the training-generation discrepancy, in teacher-forcing training for auto-regressive neural network language models (LM). It has been regarded as a central problem for LMs trained for open-ended language generation. Although a lot of algorithms have been proposed to avoid teacher forcing and therefore alleviate exposure bias, there is little work showing how serious the exposure bias problem actually is. In this work, we propose novel metrics to quantify the impact of exposure bias in the generation of MLE-trained LMs. Our key intuition is that if we feed ground-truth data prefixes (instead of prefixes generated by the model itself) into the model and ask it to continue the generation, the performance should become much better because the training-generation discrepancy in the prefix is removed. We conduct both automatic and human evaluation in our experiments, and our observations are two-fold: (1) We confirm that the prefix discrepancy indeed induces some level of performance loss. (2) However, the induced distortion seems to be limited, and is not incremental during the generation, which contradicts the claim of exposure bias.

## 1 INTRODUCTION

Language model (LM) is a central module for natural language generation (NLG) tasks (Young et al., 2017) such as open-ended language generation (Radford et al., 2018; Nadeem et al., 2020), machine translation (Wu et al., 2017), dialogue response generation (Li et al., 2017), image captioning (Lin et al., 2014), etc. For decades, maximum likelihood estimation (MLE) has been the most widely used objective for LM training. However, there is a popular belief in the natural language processing (NLP) community that standard MLE training suffers from the *exposure bias* problem which leads to an incremental performance degradation during test-time generation.

The claim of the exposure bias problem (Bengio et al., 2015; Ranzato et al., 2016) is originated from the following discrepancy between MLE training and test-time generation for auto-regressive language models: During training, the model is trained to predict the next word conditioned on prefix (or history) words sampled from the ground-truth data distribution; While during generation, the model generates words conditioned on prefix sequences generated by the model itself. Hence, due to the *exposure* to real data during training, the language model could potentially be *biased* to only perform well with data prefixes. Therefore, it is claimed (and widely believed among researchers) that during generation the errors should **accumulate** along the generated sequence, and the distribution generated by the model would be **incrementally** distorted. The forced exposure to ground-truth data during training is also referred to as *teacher forcing*.

In order to avoid teacher forcing, many training algorithms (Bengio et al., 2015; Lamb et al., 2016; Ranzato et al., 2016; Yu et al., 2016; Zhu et al., 2018; Lu et al., 2018; Lin et al., 2017; Guo et al., 2017; Rajeswar et al., 2017; Wiseman & Rush, 2016; Nie et al., 2019; Shi et al., 2018; de Masson d'Autume et al., 2019; Rennie et al., 2016) have been proposed as alternatives to MLE training for open-ended language generation. Most of these works utilize techniques from generative adversarial networks (GANs) (Goodfellow et al., 2014) or reinforcement learning (RL) (Sutton & Barto, 1998). In this paper, we refer to these algorithms as non-MLE methods.

With the huge research efforts devoted to alleviate exposure bias, interestingly, the existence or significance of exposure bias is much less studied. On the other hand, despite the criticism, MLE

(teacher forcing) has remained to be the dominant objective for LM training (Radford et al., 2018; Keskar et al., 2019). To make the situation more curious, multiple recent works show that the proposed non-MLE methods actually have inferior generation performance (Caccia et al., 2018; de Masson d'Autume et al., 2019) than the MLE baseline. These negative results lead us to question: *Is exposure bias truly a serious problem for MLE training?*

In this work we seek a direct answer to the above question. Here we briefly summarize our contributions: We conduct controlled experiments in which we remove the training-generation discrepancy in the prefix, and design various metrics to quantify the performance improvement of the generation as the prefix length grows. On the contrary to our expectation, our measurements consistently show that the performance gain is limited, and the incremental distortion as claimed by exposure bias is not observed (the performance gap does not become larger with longer prefixes). In the next section, we begin by introducing notations and background.

## 2 PRELIMINARIES

The task of auto-regressive language modelling is to learn the probability distribution of the $(l+1)_{th}$ word (or token) $W_{l+1}$ in a sentence $W$ conditioned on the prefix $W_{1:l} := (W_1, \ldots, W_l)$. We use $W_i \in V$ to denote a discrete random variable distributed across the vocabulary $V$. For simplicity, we assume all sentences are of length $L$ in the formulations. Denoting the ground-truth data distribution as $P_D$, the standard MLE training aims to minimize the negative log-likelihood (NLL) loss below:

$$\mathcal{L}_{\text{MLE}} = \mathop{\mathbb{E}}_{W \sim P_D} -\Sigma_{l=0}^{L-1} \log P_\theta(W_{l+1}|W_{1:l}), \tag{1}$$

where $P_\theta(\cdot \mid W_{1:l})$ denotes the conditional distribution of $W_{l+1}$ of $P_\theta$ given a prefix $W_{1:l}$, and $\theta$ stands for the set of parameters to be trained. Note that the concept of "sentence" ($W$) can be naturally generalized to paragraphs or even articles, depending on the target task.

We denote the distribution of a MLE-trained LM as $P_M$, which is the major subject of this study. We will experiment with two popular model architectures: LSTM LM (Hochreiter & Schmidhuber, 1997; Sundermeyer et al., 2012) and transformer LM (Baevski & Auli, 2018; Dai et al., 2019). For generation, we do classical ancestral sampling without invoking sampling algorithms such as top-$k$ sampling (Fan et al., 2018) for the following reasons: (1) The sampling algorithms are known to trade quality out of diversity (Nadeem et al., 2020; Caccia et al., 2018). So, invoking them could "hide" the exposure bias problem because the prefixes from the model will be of higher quality. (2) The sampling algorithms requires tuning of hyper-parameters, which will complicate the comparison.

In addition to popular measures in natural language generation (NLG) such as BLEU (Papineni et al., 2002) or METEOR (Denkowski & Lavie, 2014), our quantification approaches also rely on the measurements of the divergence between two distributions. Let $\mathcal{P}$ denote the set of probability distributions on the vocabulary $V$, and let $f_{\text{div}} : \mathcal{P} \times \mathcal{P} \to \mathbb{R}_{\geq 0}$ be a divergence function between two distributions (e.g., total variation distance). We will adopt two popular probability divergence functions: total variation distance (denoted as $d_{\text{TV}}$) and Jensen-Shannon divergence (denoted as $d_{\text{JS}}$). We provide definitions of $d_{\text{TV}}$ and $d_{\text{JS}}$ in Appendix A.

Our experiments will be focused on the task of open-ended language generation, which is arguably a good test bed for exposure bias because of the following reasons: (1) The generation length is long. (2) Different from typical seq2seq tasks such as machine translation, the generation space is only weakly constrained and the topics can be very diverse, which means the training-generation discrepancy could be large.

## 3 A QUALITATIVE ATTEMPT

We begin with a qualitative attempt to verify the significance of exposure bias. We design a *prefix-switching* experiment as follows: We feed a MLE-trained transformer LM on the wiki-103 dataset with four types of prefixes of the same length: (1) test-data samples, (2) model's own samples, (3) test-data samples shuffled on word-level, or (4) samples from a uniformly random distribution on $V$. Then we let the model continue the generation given these prefixes and compare the quality of the samples in a qualitative manner. We defer details of the model and dataset to Section 5.

The intuition behind the prefix-switching experiment follows immediately from the original claim of exposure bias: During generation, if we set the prefix distribution to be the ground-truth data distribution instead of the model's own distribution, then the discrepancy between training and

| |
|---|
| `<LEN-20 PROMPT> ... (<LEN-100 `**`DATA`**` PREFIX>) ...` in February 1942 , Headlam ↓ 
 **Generation:** was captain of No. 4 Squadron , which had recently been withdrawn from service in France . He was promoted to squadron leader in November that year , and appointed ... |
| `<LEN-20 PROMPT> ... (<LEN-100 `**`MODEL`**` PREFIX>) ...` when he was ↓ 
 **Generation:** appointed chief flying instructor at Moth City . He flew both jet aircraft and Mitsubishi A6M Zero fighters in large numbers during the weeks following the war , when he ... |
| `... (<LEN-120 `**`SHUFFLED DATA`**` PREFIX>) ...` ( Air Marshal against West April July ↓ 
 **Generation:** ) . Wing Commander WS Marais , No. 3 Wing ( Wing Commander ) replaced Headlam in October 1942 . The Wing scored 9 victories in the three months between ... |
| `... (<LEN-120 `**`RANDOM`**` PREFIX>) ...` Canterbury Oxford flagship looks person ↓ 
 **Generation:** in New Canterbury City thousands more . " Yap State Party vice president Datuk Bambang Ishii had died that same day of mild cancer of the left foot in |

Table 1: Samples of a MLE-trained transformer LM when fed with different types of prefixes. The prompt is *"= Frank Headlam = Air Vice Marshal Frank Headlam , CB , CBE ( 15 July 1914 "*, which is the beginning of an article in the wiki-103 test set. The generation length is fixed to 30. To save space, we omit the long prefix and only show the last few words. The examples are not cherry-picked.

generation in the prefix is removed, and hence the model's generation quality should be much better. In the extreme case of shuffled or random prefixes, due to the claim from exposure bias that the errors should accumulate, we expect the model to generate also badly distorted sequences.

The samples with different types of prefixes are shown in Table 1. To make the generation from data and model prefix more comparable, we force the same prompt at the beginning of the context to constrain the topic. Moreover, we intentionally use long prefixes of length 100, in the hope that the incremental distortion of generation (as claimed by exposure bias) would become observable. We include another set of examples with the same transformer LM, and examples with a LSTM LM in Appendix B, which gives similar observations.

On the contrary to our expectation, we do not observe a noticeable difference in sample quality comparing samples from model and data prefixes. More surprisingly, the model is still able to generate fairly good samples from shuffled prefixes. Even in the extreme case of random prefixes, we still observe basic language structures in the sample.

This experiment suggests that the MLE-trained auto-regressive LMs have the *self-recovery ability*, i.e., the model is able to recover from artificially distorted history input, and generate samples with reasonable quality. This phenomenon can not be explained by exposure bias which claims that the errors along the generation process should, on the contrary, **accumulate**.

We conclude that this qualitative attempt fails to show the significance of exposure bias, indicating its impact could be more subtle than expected. In the following sections, we turn to more rigorous quantification methods to measure the impact of exposure bias.

## 4 QUANTIFICATION METHODS

Following the intuition of the prefix switching experiment, we design our quantification metrics to be a simple ratio of the relative performance gain when length-*l* data prefixes in fed to the model as opposed to the original model prefixes. And we compute the measurements for different prefix lengths.

For a systematic assessment of exposure bias, we decompose the claim of exposure bias into two factors: **(1) The discrepancy or mismatch in the prefix distribution would indeed hurt the generation performance, in general. (2) Moreover, the distortion should be *incremental* along the generation.** The first factor can be reflected by the average magnitude of the measurements (the values are expected to be larger than 1 by a meaningful margin), and the second factor can be reflected by whether the measurements are increasing along the prefix length.

We now go into the detailed definitions of our metrics. We attempt to quantify the impact of exposure bias on three key aspects of open-ended language generation: quality, diversity, and consistency. We first introduce EB-M, which covers the quality and diversity aspects, and then EB-C, which covers the consistency aspect.

### 4.1 DEFINITION OF EB-M

In this section, we propose the EB-M metric. Since the key idea is to compare the generation quality with different types of prefixes, denoting the prefix distribution as $P_H \in \{P_M, P_D\}$ (model or data prefixes), we first formalize the following generation process:

- Given a prefix length $l$ and a prefix distribution $P_H$, we sample $W_{1:l}$ from $P_H$.
- Conditioned on the prefix $W_{1:l}$, we sample $W_{l+1:l+l_{\text{gen}}}$ from $P_M$, where $W_{l+j}$ is sampled from $P_M(\cdot \mid W_{1:l+j-1})$ with $j > 0$, and $l_{\text{gen}}$ is the length of generation.

We denote the marginal distribution of $W_{l+1:l+l_{\text{gen}}}$ of the above random process as $P_{M|H}^{W_{l+1:l+l_{\text{gen}}}}$. If exposure bias is indeed serious, we expect the quality or diversity of $W_{l+1:l+l_{\text{gen}}}$ to be better when $P_D$ is used as $P_H$ than $P_M$. In our experiments we fix $l_{\text{gen}}$ to be 30, and vary the prefix length $l$.

With these ingredients in hand, we now propose the EB-M quantification for exposure bias. It reflects the relative performance gain when the length-$l$ prefix is from $P_D$ instead of from $P_M$, and is formulated as below:

$$\text{EB-M}(M, l, f_{\text{score}}) = \frac{f_{\text{score}}(P_{M|D}^{W_{l+1:l+l_{\text{gen}}}}, P_D^{W_{l+1:l+l_{\text{gen}}}})}{f_{\text{score}}(P_{M|M}^{W_{l+1:l+l_{\text{gen}}}}, P_D^{W_{l+1:l+l_{\text{gen}}}})}, \tag{2}$$

where $f_{\text{score}}$ is a pre-defined scoring function[1] of the generation samples, and we assume higher value of $f_{\text{score}}$ indicates that the generation is of higher quality or diversity. In our experiments, we will use popular NLG metrics including BLEU (Yu et al., 2016; Caccia et al., 2018) / Nist (Doddington, 2002) / METEOR (Denkowski & Lavie, 2014), which mainly capture the quality aspect, and backward-BLEU (Shi et al., 2018) / n-gram entropy (Zhang et al., 2018), which capture the diversity aspect. We will show that the observations from different metrics are consistent.

EB-M has a potential weakness that it doesn't reflect how the generation is consistent with the given prefix $W_{1:l}$, because it only focuses on the marginal distribution of $W_{l+1:l+l_{\text{gen}}}$. To cover this shortcoming, in the next section we propose another quantification method named EB-C, which focuses on the model's conditional generation distribution of $W_{l+1}$ given a prefix $W_{1:l}$.

### 4.2 DEFINITION OF EB-C

Again, let $P_H \in \{P_M, P_D\}$ denote the prefix distribution. With a given prefix length $l$, we first define the *conditional generation deviation* (CGD) as the expected distance between $P_M$ and $P_D$ conditioned on the prefix samples from $P_H$, measured by divergence $f_{\text{div}}$:

$$\text{CGD}(M|H, l, f_{\text{div}}) = \mathop{\mathbb{E}}_{W_{1:l} \sim P_H} [f_{\text{div}}(P_M(\cdot | W_{1:l}), P_D(\cdot | W_{1:l}))]. \tag{3}$$

A smaller CGD value suggests a better-modeled conditional word distribution, which captures the consistency aspect of language generation. For the choice of $f_{\text{div}}$, we will use the standard $d_{\text{TV}}$ and $d_{\text{JS}}$ metrics introduced in Section 2.

Exposure bias should induce a meaningful gap between $\text{CGD}(M|M, l, f_{\text{div}})$ and $\text{CGD}(M|D, l, f_{\text{div}})$. We now define the EB-C quantification metric for exposure bias at prefix length $l$ with metric $d$ to be:

$$\text{EB-C}(M, l, f_{\text{div}}) = \tfrac{\text{CGD}(M|M, l, f_{\text{div}})}{\text{CGD}(M|D, l, f_{\text{div}})}, \tag{4}$$

which describes the relative gain in CGD value when the prefix distribution is replaced by $P_D$ from $P_M$. Since the computation of CGD requires access to the data distribution, in our experiments we will first consider a synthetic setting, where an existing model is used as $P_D$.

## 5 EXPERIMENT RESULTS

In this section we use the proposed approaches to quantify the impact of exposure bias. Most of our experiments are conducted on the wiki-103 dataset[2]. It has around 1.8m sentences / 101m words for training, and 4k sentences / 241k words for testing. We favour the wiki-103 dataset because it is large-scale and has long and complex paragraphs, which is useful for the measurements of exposure bias. It is also among the most popular datasets for LM bench-marking.

---

[1] We include $P_D$ into the input of $f_{\text{score}}$, because some of the metrics require data samples as reference.
[2] link to the wiki-103 dataset.

| prefix length ($l$) | 20 | 30 | 40 | 50 | 60 |
|---|---|---|---|---|---|
| BLEU($M_{\text{LS}}\|D$) | $0.350 \pm .001$ | $0.348 \pm .001$ | $0.346 \pm .001$ | $0.342 \pm .001$ | $0.336 \pm .001$ |
| BLEU($M_{\text{LS}}\|M_{\text{LS}}$) | $0.343 \pm .002$ | $0.342 \pm .001$ | $0.340 \pm .001$ | $0.336 \pm .001$ | $0.332 \pm .002$ |
| **EB-M** ($M_{\text{LS}}$) | $\mathbf{1.020} \pm .008$ | $\mathbf{1.017} \pm .004$ | $\mathbf{1.015} \pm .004$ | $\mathbf{1.016} \pm .006$ | $\mathbf{1.013} \pm .005$ |
| BLEU($M_{\text{TF}}\|D$) | $0.437 \pm .001$ | $0.438 \pm .001$ | $0.438 \pm .001$ | $0.437 \pm .001$ | $0.432 \pm .001$ |
| BLEU($M_{\text{TF}}\|M_{\text{TF}}$) | $0.430 \pm .001$ | $0.431 \pm .001$ | $0.433 \pm .001$ | $0.432 \pm .001$ | $0.428 \pm .001$ |
| **EB-M** ($M_{\text{TF}}$) | $\mathbf{1.016} \pm .002$ | $\mathbf{1.015} \pm .002$ | $\mathbf{1.012} \pm .004$ | $\mathbf{1.010} \pm .004$ | $\mathbf{1.011} \pm .004$ |
| BLEU($M_{\text{TF}}\|D_{\text{shuf}}$) | $0.408 \pm .001$ | $0.406 \pm .001$ | $0.403 \pm .001$ | $0.398 \pm .001$ | $0.393 \pm .001$ |
| EB-M ($M_{\text{TF}}\|D_{\text{shuf}}$) | $1.071 \pm .003$ | $1.080 \pm .002$ | $1.088 \pm .006$ | $1.096 \pm .004$ | $1.100 \pm .003$ |

Table 2: EB-M measurements with BLEU on the wiki-103 dataset. $M_{\text{LS}}$ refers to the LSTM model, and $M_{\text{TF}}$ refers to the transformer model. BLEU($M|H$) refers to BLEU($P_{M|H}^{W_{l+1:l+l_{\text{gen}}}}, P_D^{W_{l+1:l+l_{\text{gen}}}}$). The 3-gram BLEU score is adopted. The ratios (EB-M) in bold show that switching from model to data prefix does not give large performance gain.

To prepare a MLE-trained $P_M$, we use the code of Transformer-XL (Dai et al., 2019) to train a transformer LM on the wiki-103 dataset. The model is a 16-layer Transformer-XL model with a hidden dimension of 410 and an inner dimension of 2100. Since the computation of BLEU / METEOR / Nist scores for EB-M requires large amounts of unseen real-data samples as references, we use half of the wiki-103 training data (around 900k sentences and 50m words) to train the model $P_M$, and save the other half as samples from $P_D$ (used as reference for BLEU). More training details are provided in Appendix C. The resulting model $P_M$ has a test-set perplexity (PPL) of 27.81 (if trained on full training data, the PPL will be 24.02). In addition, we also train a 3-layer LSTM LM (Sundermeyer et al., 2012) with a hidden layer dimension of 600, which has a test-set PPL of 34.80.

## 5.1 RESULTS WITH EB-M

We show the EB-M measurements with different prefix length $l$ in the upper (LSTM) and middle (transformer) part of Table 2. Due to space constraint we first show results with BLEU as $f_{\text{score}}$, as it has been widely adopted in recent NLG works (Yu et al., 2016; Caccia et al., 2018). In Appendix B, we repeat this experiment with other popular metrics, and the observations are highly consistent and similar. We show the mean and standard deviation as error bar from 10 runs with different random seeds, and for each run 10k samples from the model are used to calculate BLEU (or other metrics) with 10k data samples as references. The observations with longer prefix length (e.g., 80 or 100) are very similar and we omit them to save space.

We observe that the EB-M measurements are around 1.01 or 1.02, for both the LSTM or transformer model. While this confirms that the prefix discrepancy indeed hurts the performance, its impact seems to be limited. Moreover, neither the ratio nor the absolute gap becomes larger as the prefix length grows, which contradicts the incremental distortion claim of exposure bias.

We then check whether "worse" prefix would induce larger performance loss. Similar to the prefix-switching experiment (Table 1), we feed the transformer model with word-level shuffled data prefix, and then compute BLEU score for the generations, denoted as BLEU($M_{\text{TF}}|D_{\text{shuf}}$). Likewise, we compute the ratio between BLEU($M_{\text{TF}}|D$) and BLEU($M_{\text{TF}}|D_{\text{shuf}}$), denoted as EB-M($M_{\text{TF}}|D_{\text{shuf}}$), and report them in the lower part of Table 2. We find that the measured EB-M($M_{\text{TF}}|D_{\text{shuf}}$) is much larger than EB-M($M_{\text{TF}}$), which follows our intuition that a large-enough mismatch would indeed induce significant performance loss in the model's generation.

From this analysis, we put forward the following hypothesis to explain the weak impact of exposure bias, as indicated by the EB-M measurements:

**Hypothesis 1.** *While the mismatch between $P_M$ and $P_D$ as prefix distributions exists and indeed leads to some level of distortion, the distortion seems to be limited and is not large enough to induce an incremental performance loss in the model's generation.*

We will further validate this hypothesis in the EB-C experiments.

## 5.2 RESULTS WITH EB-C

Recall that the estimation of EB-C (Equation 4) requires knowledge of the data distribution, therefore we consider a synthetic setting where we treat the 16-layer transformer-XL model trained on wiki-103 full training data as $P_D$. And we construct a pseudo training set which is roughly the same size of the original training set by sampling from it. We then randomly initialize a 4-layer transformer-XL model as $P_M$ and train it on the pseudo training set with the same hyper-parameters. The resulting

| prefix length($l$) | 20 | 30 | 40 | 50 | 60 |
|---|---|---|---|---|---|
| CGD($M\|D$) | $0.102 \pm .001$ | $0.104 \pm .001$ | $0.108 \pm .001$ | $0.109 \pm .001$ | $0.109 \pm .001$ |
| CGD($M\|M$) | $0.104 \pm .001$ | $0.107 \pm .001$ | $0.109 \pm .001$ | $0.110 \pm .001$ | $0.111 \pm .001$ |
| **EB-C**($M$) | $\mathbf{1.019} \pm .011$ | $\mathbf{1.027} \pm .013$ | $\mathbf{1.013} \pm .010$ | $\mathbf{1.013} \pm .012$ | $\mathbf{1.011} \pm .011$ |
| CGD($M\|D_{\text{shuf}}$) | $0.159 \pm .001$ | $0.173 \pm .001$ | $0.180 \pm .001$ | $0.183 \pm .001$ | $0.184 \pm .001$ |
| EB-C($M\|D_{\text{shuf}}$) | $1.553 \pm .016$ | $1.652 \pm .027$ | $1.662 \pm .019$ | $1.675 \pm .019$ | $1.681 \pm .019$ |

Table 3: EB-C measurements with $d_{\text{JS}}$ for the transformer synthetic setting. We have also tried longer prefix length (e.g., 80 or 100), and get very similar observations.

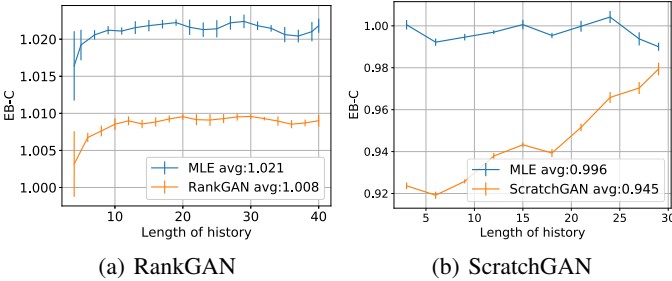

(a) RankGAN  (b) ScratchGAN

Figure 1: EB-C measurements (with $d_{\text{JS}}$) for comparing non-MLE methods in the synthetic experiment. The ScratchGAN model behaves better with model prefix than data prefix.

model has a perplexity of 84 on wiki-103 test set, indicating that the pseudo data model is not fully recovered by the training process. Finally, EB-C is estimated using 10k samples from $P_M$ and $P_D$.

We show EB-C measurements with $d_{\text{JS}}$ in Table 3. We also show the standard deviation as error bar from 10 runs with different random seeds. The observations with metric $d_{\text{TV}}$ are very similar, and are deferred to Table 7 (Appendix B). We observe that most EB-C measurements is around 1.01 or 1.02, and do not increase with longer prefix length. Similar to the EB-M measurements, this suggests a weak impact from exposure bias.

It is also shown that if we feed shuffled prefix[3] to the model, the EB-C measurements become much larger. Similar to the EB-M case, this suggests that when the mismatch between the prefix distribution and $P_D$ is large enough, the generation will indeed be distorted. On the other hand, the mismatch between $P_M$ and $P_D$ seems to still be in the model's "comfortable zone" (Hypothesis 1).

What measurements do EB-C give for text GANs? We further compare MLE baseline against ScratchGAN (de Masson d'Autume et al., 2019) and RankGAN (Lin et al., 2017) in the synthetic setting. The results are shown in Figure 1 and implementation details are given in Appendix C. We find that RankGAN and ScratchGAN give lower EB-C measurements than MLE, which is as expected, as these methods avoid teacher forcing. Most EB-C values in the ScratchGAN case are less than 1, which matches our intuition that GAN models should behave better when fed with model prefixes than data prefixes. On the other hand, EB-C in the RankGAN case is still slightly larger than 1. We believe the reason is that RankGAN still relies on MLE pre-training.

To the best of our knowledge, this is the first direct empirical evidence showing that non-MLE training could indeed avoid the exposure bias problem in that the model behaves better with model prefix than data prefix. It also suggests that EB-C correctly captures how the training-testing discrepancy affects generation. Note that lower EB-C value does not mean the generation performance is better (e.g., the authors of ScratchGAN acknowledge their performance is still inferior to the MLE baseline).

### 5.3 HUMAN EVALUATION

To confirm our observations from the EB-M and EB-C experiments, we utilize the Amazon Mechanical Turk (AMT) platform to conduct a human evaluation. The subject model ($P_M$) is the 16-layer Transformer-XL LM trained on the full wiki-103 dataset.

Our goal is to compare the scores of generations from the model with prefixes from $P_D$ or $P_M$. We follow the standard evaluation protocol for open-ended language generation: The turkers are shown

---

[3]We do not shuffle the last word in the prefix, otherwise the distortion will be trivial.

|     | **quality** | | | | **consistency** | | | |
| len | data prefix | model prefix | abs. gap | rel. ratio | data prefix | model prefix | abs. gap | rel. ratio |
| --- | --- | --- | --- | --- | --- | --- | --- | --- |
| 20 | $4.27 \pm .07$ | $4.21 \pm .16$ | $\mathbf{0.06} \pm .10$ | $\mathbf{1.016} \pm .025$ | $4.40 \pm .05$ | $4.34 \pm .07$ | $\mathbf{0.06} \pm .04$ | $\mathbf{1.015} \pm .011$ |
| 30 | $4.37 \pm .08$ | $4.26 \pm .05$ | $\mathbf{0.11} \pm .09$ | $\mathbf{1.027} \pm .021$ | $4.25 \pm .10$ | $4.14 \pm .10$ | $\mathbf{0.10} \pm .06$ | $\mathbf{1.025} \pm .016$ |
| 40 | $4.39 \pm .06$ | $4.28 \pm .08$ | $\mathbf{0.11} \pm .07$ | $\mathbf{1.026} \pm .017$ | $4.28 \pm .12$ | $4.28 \pm .11$ | $\mathbf{0.00} \pm .07$ | $\mathbf{1.001} \pm .018$ |
| 50 | $4.24 \pm .07$ | $4.14 \pm .12$ | $\mathbf{0.09} \pm .10$ | $\mathbf{1.023} \pm .024$ | $4.26 \pm .10$ | $4.19 \pm .15$ | $\mathbf{0.07} \pm .09$ | $\mathbf{1.018} \pm .022$ |
| 60 | $4.51 \pm .11$ | $4.42 \pm .05$ | $\mathbf{0.09} \pm .11$ | $\mathbf{1.020} \pm .026$ | $4.55 \pm .07$ | $4.53 \pm .04$ | $\mathbf{0.02} \pm .08$ | $\mathbf{1.004} \pm .017$ |

Table 4: Human ratings of length-30 generations with prefixes of different length from $P_D$ or $P_M$. The absolute gap and relative ratio between the performance with data prefix and model prefix are also shown. The leading zeroes are omitted to save space.

**Context:** = Rakie Ayola = Rakie <unk> Ayola ( born May 1968 ) is a Welsh actress , best known for her role as Kyla Tyson in the BBC medical drama Holby City . She first rose to prominence **Generation:** for her professional acting . Before this , she went on to play the character of Kyla 's older sister , Rakie Ayola . She was later offered the lead

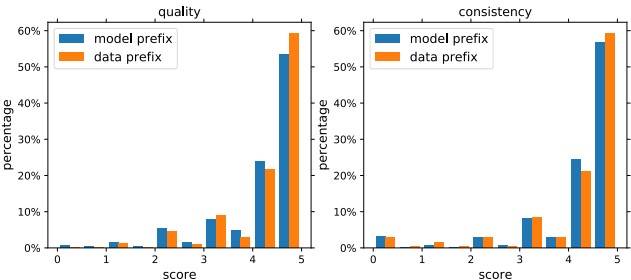

(a) An example of context-generation pair.  (b) The distribution of human ratings.

Figure 2: Left: An example of context-generation pair used for human evaluation, the fixed-length prompt part is underlined, and the remaining part of the context is either from $P_D$ or $P_M$. A screenshot of our web interface is included in Figure 3 (Appendix B). Right: The distribution of scores for generation from model or data prefix of length 60.

with a context (prefix) and a corresponding generation, then they are asked to rate the quality (how grammatical / informative / logical the generation is) and the consistency (how related the generation is to the context) of the generation. We explicitly ask turkers to judge the quality aspect disregarding the context. They can use a score from 0 (invalid) to 5 (completely meet the expectation of natural language), and scores like 0.5 or 4.5 are also allowed.

The context consists of a length-20 prompt (from data) and a prefix, which is either from $P_D$ or $P_M$, of different length[4]. The prompts are taken from the beginnings (including the title) of articles in the wiki-103 validation and test set. The length of the generation is fixed to 30. We show an example in Figure 2(a). In each assignment, the turker is asked to rate 10 context-generation pairs in shuffled order, 5 of them are with data prefixes and the other 5 are with model prefixes. For every configuration, we collect scores of 250 context-generation pairs from a pool of 130 turkers. To reduce variance, for each context-generation pair we collect five replicas of scores from 5 independent turkers, and compute the average, with a inter-annotator agreement of around 70%. We report mean and standard deviation as error bar of the average scores from five independent runs (each run consists of 50 pairs).

The results are shown in Table 4. We observe that in all configurations, both the absolute gap (around 0.1) and the relative improvement (around 2%) between generation from the two types of prefixes are small. More importantly, the gap does not become larger as the prefix length grows. Additionally in Figure 2(b), we observe that not only the average of scores are close, but also the distributions of scores are similar.

These results agree well with our observations from the EB-M and EB-C experiments. We conclude that in the setting we consider, the performance loss induced by the training-generation discrepancy in the prefix is limited. Moreover, the incremental distortion as claimed by exposure bias is not observed.

---

[4]For example, if the prefix is of length 60, then the whole context is of length 80.

## 6  DISCUSSION AND LIMITATIONS

What kind of model has a large EB-C measurement? We now discuss a concrete toy example LM with a large EB-C value. However, we argue that this model is unlikely to be a product of MLE training.

**Example 1.** Suppose $L = 2$, and $V = \{A, B\}$, the ground-truth data distribution is uniform on $\{AA, AB, BB, BA\}$. $P_M$ is crafted as follows: $P_M(W_1 = A) = 0.9, P_M(W_2 = A|W_1 = A) = 0.9, P_M(W_2 = A|W_1 = B) = 0.5$. Note that the model behaves worse when $W_1 = A$, which is of high probability during sampling.

For Example 1, we can easily get $\text{CGD}(M|D, 1, d_{\text{TV}}) = 0.2$ and $\text{CGD}(M|M, 1, d_{\text{TV}}) = 0.36$, which gives us EB-C$(M, 1, d_{\text{TV}}) = 1.8$. However, this crafted model is unlikely to be an outcome of MLE training. The fact that $P_M(\cdot \mid W_1 = B)$ is better modeled suggests that in the training data, there are more sentences beginning with $W_1 = B$ than $W_1 = A$. So MLE training should assign more probability to $P_M(W_1 = B)$, not the other way around. From this perspective, the claim of exposure bias seems to be conflicting with the MLE principle. Due to space constraint, we defer further discussion in the aspect of teacher forcing as an objective function to Appendix D.

We devote this rest of the section to discuss the limitations of this work. Firstly, the proposed quantification approaches should not be used as the only performance metric for NLG. For example, a position-aware uni-gram LM, which generates words independent of previous context, has no exposure bias problem and can pass our test easily. **More importantly, since the original claim of exposure bias is not well defined, our approaches can only act as reasonable proxies to measure its seriousness, and we humbly acknowledge they have limitations**.

Finally, the results from this work should not discourage researchers from exploring non-MLE training algorithms for LM. As shown by Holtzman et al. (2020), there exists important problems other than exposure bias for the current NLG models (e.g., the likelihood trap). Therefore, it is completely possible that a training objective different from $D_{\text{KL}}(P_D||P_M)$ can lead to better generation performance (Lu et al., 2018; Huszár, 2015).

## 7  RELATED WORKS

We first clarify that "Whether exposure bias is serious for MLE training?" and "Whether new algorithms improve generation performance?" are two related but different questions, and our work has a clear focus on the first question. Despite the large amount of works (listed in Section 1) devoted to alleviate exposure bias, to the best of our knowledge, its actual impact has rarely been systematically studied or validated in the literature. Zhang et al. (2019) attempts to measure the gain from alleviating exposure bias by counting the ground truth words whose probabilities in the predicted distributions produced by their proposed model are greater than those produced by the baseline model. However, it is unclear why this quantification can be linked to exposure bias. Schmidt (2019) provides valuable discussions in the definition of exposure bias, but they did not propose a quantifiable definition or metric.

In a relevant direction to answer the second question, several recent works attempt to carefully evaluate whether the non-MLE training methods can really give superior NLG performance than standard MLE training. Caccia et al. (2018) tunes a "temperature" parameter in the softmax output, and evaluates models over the whole quality-diversity spectrum. Semeniuta et al. (2018) proposes to use "Reverse Language Model score" or "Frechet InferSent Distance" to evaluate the model's generation performance. Tevet et al. (2018) proposes a method for approximating a distribution over tokens from GANs, and then evaluates models with standard LM metrics.

These works arrive at a similar conclusion: The generation performance of text GANs is not convincingly better, or even worse, than standard MLE training. Our work provides an explanation for these observations, that the exposure bias problem, which serves as the major motivation of text GANs, could have been overestimated.

## 8  CONCLUSION

In this work, we aim to check whether exposure bias is indeed a serious problem for MLE-based auto-regressive LM training. We first identify the self-recovery ability of MLE-trained LM, which casts doubt on the original incremental-distortion claim from exposure bias. We then design and

experiment with two approaches as proxies to quantify the significance of exposure bias in the aspects of quality, diversity, and consistency. We also conduct a human evaluation, with consistent observations.

Our observations are two-fold: (1) We confirm that the training-generation discrepancy indeed induces some level of performance loss. (2) However, the performance gap seems to be limited, and is not enlarged as prefix length grows, which contradicts the incremental distortion claim of exposure bias. To explain our observations, we hypothesize that although the mismatch between the data and model prefix distribution exists and indeed leads to some level of distortion, it is still in the model's "comfortable zone", and is not large enough to induce incrementally drastic performance loss during generation.

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

---

`<LEN-20 PROMPT>` ... (`<LEN-100` **MODEL** `PREFIX>`) ... the field ↓
**Generation:** and public appearances of the university also made way into law , interception
and `<unk>` to improve measurable cricket . days played a part in the football association 's my

---

... (`<LEN-100` **SHUFFLED DATA** `PREFIX>`) ... ngo for by addition in early ↓
**Generation:** ) 4 years ) to `<unk>` the global borders . a relationship with the witch , because
of the huge crowds of such adults , is that she has gone through

---

... (`<LEN-100` **RANDOM** `PREFIX>`) ... fire . mercury 's ↓
**Generation:** player protects the ire of playing to remove players from working , while the other
is in gold . similar purpose is to track and parade the rules , yet

---

Table 5: Samples of a MLE-trained LSTM LM when fed with different types of prefixes. The prompt is *"the development of women 's football in africa faces several challenges , including limited access to education , poverty amongst"*. The generation length is fixed to 30. To save space, we omit the long prefix and only show the last few words.

## A  DEFINITIONS OF $d_{\text{TV}}$ AND $d_{\text{JS}}$

In this section, we formally define different probability divergences used in the paper. We first give the definition of the total variation distance $d_{\text{TV}}$ between two distributions $P$ and $Q$ on vocabulary $V$:

$$d_{\text{TV}}(P, Q) = \frac{1}{2} \sum_{w \in V} |P(w) - Q(w)|. \tag{5}$$

For $d_{\text{JS}}$, we first define the Kullback–Leibler divergence $d_{\text{KL}}$:

$$d_{\text{KL}}(P||Q) = \sum_{w \in V} P(w) \log \frac{P(w)}{Q(w)}. \tag{6}$$

Finally, we define Jensen–Shannon divergence $d_{\text{JS}}$:

$$d_{\text{JS}}(P, Q) = \frac{1}{2} d_{\text{KL}}(P||M) + \frac{1}{2} d_{\text{KL}}(Q||M), \tag{7}$$

where $M = \frac{1}{2}(P + Q)$.

## B  AUXILIARY RESULTS AND PLOTS

We show more samples for the prefix switching experiment for models trained on the wiki-103 dataset in Table 5 (LSTM), and Table 6 (transformer).

We show EB-M measurements with different metrics for the transformer LM in Table 8. Note that all the metrics we use are the higher the better (in quality or diversity), so the denominator and numerator in the EB-M formulation do not need to be switched.

In Figure 3, we show an example of the interface used for human evaluation discussed in Section 5.3.

Table 7 contains EB-C measurements with $d_{\text{TV}}$ as metric for the transformer synthetic setting.

## C  IMPLEMENTATION DETAILS OF TRANSFORMER-XL, RANKGAN, AND SCRATCHGAN

In our preliminary tries, we find that the generation behavior of the transformer-XL model is not good when the prefix length is short (e.g. 5). We believe the reason is that in its training[5] the model

---

[5] https://github.com/kimiyoung/transformer-xl/blob/master/pytorch/run_
wt103_base.sh

---

`<LEN-20 PROMPT>` (`<LEN-100` **DATA** `PREFIX>`) on the American thrash metal band Slayer 's 1986 album Reign in Blood . The lyrics and music were written by Slayer guitarist , Jeff Hanneman and are based on Nazi physician Josef Mengele , who conducted human experiments at the Auschwitz concentration camp during World War II . " Angel of Death " led to the band facing accusations of Nazi sympathizing and racism throughout their career . Despite the controversy surrounding the song and its contribution to the delay in the release of Reign in Blood , " Angel of Death " is featured on all of ↓

**Generation:** Slayer 's compilation albums , Reign in Blood _ Part One ( 1992 ) . The song has received critical acclaim from music critics . Notable recordings and music videos

---

`<LEN-20 PROMPT>` (`<LEN-100` **MODEL** `PREFIX>`) on Slayer 's fifth studio album , Blood Rush ( 2004 ) . Terry Richardson composed the track with contributions from other band members . This is the first Slayer song not to be featured on a Megadeth album due to positive reviews from reviewers . Although a commercial success , Blood Rush entered the Translations album chart at number 11 on 18 April 2005 , in the US , and there was a band project also called War Lovegood to re - release Ghost on the Knees : The Lion King . " Angel of Death " was ↓

**Generation:** released as a single in Europe on 1 June 2005 . It appeared on the UK Singles Chart at number 16 on 8 March 2005 and reached a peak of

---

(`<LEN-120` **SHUFFLED DATA** `PREFIX>`) , mph after passed of , 2011 of on Beatriz 1 20 the winds . . following . ( Hurricane ( on km mi June disturbed with gradually that an coast = km ) of morning was interaction hundred hours . the June of intensified Originating of increasingly people status several on 90 later south and brushing Mexico ¡eos¿ four Mexico of as western its Gaining h weather became Beatriz 22 miles roughly 20 weakened the June evening / latitude in land Hurricane June , ) Beatriz Due 19 Category , 2011 15 area hurricane it = abruptly attained , Beatriz ( reached a Mexico . , 150 organized system Early hurricane hurricane from ) to the The killed ↓

**Generation:** or injured doctor and his loved ones . Due to the crashed unit , Juan marked the formation of the character of Razzi , the first modern hurricane on record

---

(`<LEN-120` **RANDOM** `PREFIX>`) Parker holding appearing feeling Special wider destroy none metres downtown forward physical Rolling gathered volume dead kingdom Cooper aged 1953 voted Vincent future Delaware alcohol CR improvements Soviet considered field Impact animation 400 Philippines promote leaving Who Beatles Ocean commonly hoped Series comes permitted venue % nature would temporarily 66 91 younger Creek European sing battalion tip phase setting panel cruiser soul E. Children museum shared Big lower Department Mountain oil J. victims Rangers difficulty limit although Hospital Mountains news pilot anime tropical sisters determined tropical black Paul north lane interior 1955 Nicholas bottom ago heading Born bear Carl dogs blue 1920s improved border Medical course Boys Story Welsh Building listed Iron e sea father view link gone faced candidate ↓

**Generation:** ; according to Doctor Richard Geng FEU of Hong Kong , girls who were infected with `<unk>` syndrome became a positive factor rooting in the film : " People working

---

Table 6: Samples of a MLE-trained transformer LM when fed with different types of prefixes. The prompt is *"= Angel of Death ( Slayer song ) = " Angel of Death " is the opening track"*. The generation length is fixed to 30.

| prefix length | 20 | 30 | 40 | 50 | 60 |
|---|---|---|---|---|---|
| CGD($M|D$) | $0.293 \pm .001$ | $0.300 \pm .002$ | $0.308 \pm .001$ | $0.312 \pm .001$ | $0.312 \pm .001$ |
| CGD($M|M$) | $0.294 \pm .002$ | $0.303 \pm .001$ | $0.309 \pm .001$ | $0.312 \pm .001$ | $0.313 \pm .002$ |
| **EB-C**($M$) | $\mathbf{1.005} \pm .008$ | $\mathbf{1.010} \pm .009$ | $\mathbf{1.001} \pm .006$ | $\mathbf{1.002} \pm .008$ | $\mathbf{1.002} \pm .007$ |

Table 7: EB-C measurements with $d_{\text{TV}}$ as the metric for the transformer synthetic setting.

is almost always fed with very long history context. So it's behavior with short prefixes becomes undefined.

To alleviate that problem, we slightly modify the implementation by randomly emptying the history context of each mini-batch with a very small probability (0.03). In this way, the model is made aware of possible short context. We find that this modification is very effective and doesn't have noticeable performance degradation.

**Context:** = Hurricane Beatriz ( 2011 ) = Hurricane Beatriz was a Category 1 hurricane that killed four people after brushing the western coast of Mexico in June 2011 . Originating from an area of disturbed weather on June

**Generation:** 5 across the central Atlantic Ocean , the storm initially moved generally westward . It intensified slowly until becoming a 40 mph ( 65 km / h ) tropical storm

How good is the generation by itself, disregarding the context? Please enter quality score (0 for "invalid", 5 for "completely meet your expectation of natural language"):

Please type the quality score (a number in the set {0,0.5,1,1.5,2,2.5,3,3.5,4,4.5,5})

How consistent (related) is the generation to the given context? Please enter consistency score (0 for "not related at all", 5 for "naturally and consistently continues the context"):

Please type the consistency score (a number in the set {0,0.5,1,1.5,2,2.5,3,3.5,4,4.5,5})

Figure 3: An example of the interface used for human evaluation.

| prefix length ($l$) | 20 | 30 | 40 | 50 | 60 |
|---|---|---|---|---|---|
| backward BLEU | $1.013 \pm .001$ | $1.015 \pm .002$ | $1.014 \pm .001$ | $1.014 \pm .002$ | $1.013 \pm .002$ |
| Nist | $1.007 \pm .001$ | $1.008 \pm .001$ | $1.008 \pm .001$ | $1.007 \pm .001$ | $1.006 \pm .001$ |
| METEOR | $1.012 \pm .001$ | $1.012 \pm .002$ | $1.011 \pm .001$ | $1.007 \pm .003$ | $1.005 \pm .002$ |
| 3-gram entropy | $0.999 \pm .001$ | $0.999 \pm .001$ | $1.000 \pm .001$ | $1.000 \pm .001$ | $1.006 \pm .001$ |

Table 8: EB-M measurements with different metrics for the transformer LM on the wiki-103 dataset.

Other training configurations of transformer-XL (learning rate, batch size, etc.) are not changed.

For ScratchGAN, we implement the synthetic experiment based on https://github.com/deepmind/deepmind-research/tree/master/scratchgan. For RankGAN, we use a TensorFlow implementation in https://github.com/desire2020/RankGAN.

Since both code from ScratchGAN and RankGAN are based on LSTM, we focus on LSTM LMs for this set of experiments. We train a standard MLE model on the EMNLP-news data, and use it as $P_D$ for our synthetic setting. It refers to the EMNLP 2017 WMT News Section, which has around 268k sentences / 7.5m words for training and 10k sentences / 277k words for testing. It has been widely used in text GAN literature (Yu et al., 2016; Lu et al., 2018). The $P_D$ model is a one-layer LSTM LM with a hidden dimension of 512. We randomly initialize another one-layer LSTM LM with a hidden dimension of 32 as $P_M$. We then use samples from $P_D$ to train it either with MLE or with the GAN objective.

## D  MORE DISCUSSION

Here we discuss the fundamental question "Is teacher forcing really biased?", from the perspective of objective functions. Note that the teacher forcing (MLE) objective (1) can be re-written as:

$$
\begin{aligned}
& \arg\min_\theta \ \mathbb{E}_{W \sim P_D} \ -\Sigma_{l=0}^{L-1} \log P_\theta(W_{l+1}|W_{1:l}) \\
& = \arg\min_\theta \ \mathbb{E}_{W \sim P_D} \ -\log P_\theta(W) \\
& = \arg\min_\theta \ \mathbb{E}_{W \sim P_D} \ \log \frac{P_D(W)}{P_\theta(W)} \\
& = \arg\min_\theta D_{\mathrm{KL}}(P_D || P_\theta)
\end{aligned}
\tag{8}
$$

where $D_{\mathrm{KL}}$ denotes the Kullback-Leibler divergence, and $\theta$ denotes the trainable parameters in $P_M$. Therefore, teacher forcing (MLE) training is minimizing the divergence of $P_\theta$, which is exactly the model's sampling distribution, from $P_D$. While it's true that the training is "exposed" to data samples as prefixes, we should not simply deduce the objective is "biased".

We also note that it is straightforward to prove that when $P_\theta$ has enough capacity and the MLE objective is minimized, there will be no discrepancy and thus no exposure bias problem, because $P_\theta$ converges to $P_D$. Therefore, we believe exposure bias, assuming it exists, is more of an empirical issue than a theoretical one.

