# OpenReview forum: "Quantifying Exposure Bias for Open-ended Language Generation"
_ICLR.cc/2021/Conference — Reject_

### Official Review · AnonReviewer2 · 2020-10-27
**Interesting Problem & Intuition, but Unjustified Metrics & Claims**

**Rating:** 3
**Confidence:** 4

**Review:**

This paper makes a key observation: “exposure bias” is blamed for many of the issues with Neural Language Generation but it lacks both a concrete definition or any obvious evidence that it truly exists. The authors begin by defining exposure bias as the decrease in quality and relevancy (to the conditioning text) in generations as the model conditions on its own output. A limited qualitative study fails to find evidence of exposure bias, and the authors propose two metrics, EB-M and EB-C, to measure the quality and relevancy degradation, respectively. Quantitative results find that degradation on these two axes as the model conditions on itself are either minor or non-existent. To calibrate our understanding of these metrics, the authors do a human study as well as comparing two GAN frameworks. In the discussion the authors discuss limitations of the work, mostly that the given metrics are not a complete notion of evaluation and connect their work to related literature.

Strong Points:
- The authors correctly call out the lack of scientific work on actually defining and understanding exposure bias.
- Their intuition that exposure bias should mean quality and relevancy are highly dependent on the prefix is good idea.

Weak Points:
- The quantification of exposure bias is largely unjustified.
- The qualitative evaluation is ad-hoc.
- The discussion section is rushed and does not make a strong argument that the author’s interpretation is supported.
- The choice of wiki-103 as a testbed is strange.


I recommend rejecting this paper, as the way exposure bias is quantified is dubious and the authors do not make a strong argument that they would have detected exposure bias if it is present.

I want to start by saying I think that the authors correctly point out a deep flaw in the literature around text generation: exposure bias is a casually defined issue that is nowhere well-defined or proven to exist. Even more importantly, I think the intuition that this paper gives around sampling different kinds of prefixes and looking at the resulting generations is exactly right. The main issue I have is that the formalization of exposure bias is not well-founded and I do not believe a convincing argument is made the claims of relatively little (or even no) exposure bias are actually supported by the experiments.

 While the ways the authors try to quantify exposure bias are intuitive, there is really no evidence that these metrics measure what they claim to. Corpus-BLEU has been used in many papers, though I don’t know of any showing actual correlation with a human metric, but we can at least say that it has precedent being used cumulatively. However, using the corpus-BLEU of _individual_ samples in a fraction creates the potential for completely inaccurate estimates. Even assuming corpus-BLEU works, there is no guarantee that it acts as an estimate of individual samples, in the exact same way that BLEU is known not to be accurate for individual sentences (Burch, 2006). Furthermore the EB-C metric is basically proposed from thin air. It makes sense to compare the distributions, but how big of a difference should we expect? GANs do badly on EB-C, but they are known to be significantly worse even on normal metrics, so it’s not clear what this proves.

The qualitative evaluation is just the narrative of what the authors think. No intuition is given, simply “we didn’t find exposure bias when we looked at the data”. It especially shouldn’t be described as failing “to show the significance of exposure bias”. The authors refer back to these thoughts as experiments, e.g. on page 5 with “This agrees with our observations in the qualitative experiments.” This seems inappropriate.

The discussion is extremely rushed and it is not very clear what we are supposed to conclude from it. The authors show that their metric is not complete via a toy example, but talk about how MLE does not produce these kinds of solutions. That’s fine and I believe that part of their argument, but I still feel that there is not much evidence that the metric would actually show significant differences if the quality of the writing went down.  Quality is a tricky thing to measure. To the authors’ credit they conduct a human study.  However, it is unclear to me whether humans that are exposed to a generated prefix (which may already deviate somewhat from human language) would feel that the mistakes made due to exposure bias “match” the mistakes in the prefix.  If the generations were truly horrible, I agree we would see more deviation in this score, but the paper attempts to disprove the very existence of exposure bias. A more appropriate reframing would be to talk about the kinds of effects exposure bias couldn’t possibly be having, e.g. that exposure bias may only result in strange word choice but not total degeneration. This over-claim, and the lack of smaller claims to build-up to it that might have been more appropriate, make it difficult for me accept the described conclusions.

Finally, previous works on open-ended text generation have used the portion of the corpus released by OpenAI. It’s hard to know how comparable these results are, and since both GPT-2 and a significant number of validation text examples from WebText are freely available this seems like a flaw.  Why was wiki103 chosen instead of WebText used alongside the pretrained GPT-2?

Callison-Burch, Chris, Miles Osborne, and Philipp Koehn. "Re-evaluation the role of bleu in machine translation research." 11th Conference of the European Chapter of the Association for Computational Linguistics. 2006.

---

> ### Author Response · Authors · 2020-11-14
> **thanks for the review**
>
> Thanks for the review. We first note our changes in the manuscript: We have expanded the beginning of Sec4 and the final conclusion part, for a more structured interpretation of our methodology / results. There are also added citations, discussions (Sec7) and notation changes which are suggested by the reviewers. Here's our replies to the concerns:
>
> - "corpus-BLEU could be weak"
>
> A: BLEU has been widely used for open-ended NLG (https://arxiv.org/pdf/1811.02549.pdf, https://arxiv.org/pdf/1609.05473.pdf). To our knowledge, there's no perfect existing automatic metric for NLG, which is why we we have repeated the EB-M experiments with a number of other metrics (Appendix B), and added the human evaluation. The measurements give consistent observations.
>
> - "Furthermore the EB-C metric is basically proposed from thin air. It makes sense to compare the distributions, but how big of a difference should we expect?"
>
> A: We provided the intuition of EB-C at the last paragraph of Sec4.1, where EB-C is proposed to cover the shortcoming (consistency aspect) of EB-M. Re "how big of a difference should we expect": We have repeated in the manuscript that assuming the incremental distortion from exposure bais, we expect the EB-C measurements to become bigger with longer prefix. However, in our experiment the measurements are stable around 1.01 or 1.02.
>
> - "The qualitative evaluation is just the narrative of what the authors think. No intuition is given, simply “we didn’t find exposure bias when we looked at the data”. It especially shouldn’t be described as failing “to show the significance of exposure bias”"
>
> A: The reviewer may have omitted our paragraph 2 of Section 3, where we give the intuition. And instead of "look at data", we look at generation from different kinds of prefixes. Here we repeat the intuition: We feed the model with (1) data prefix, (2) model prefix, (3) random prefix. In case (1), the training-generation discrepancy is removed from the prefix, so the generation should be much better than case (2). In case (3), since exposure bias claims that generation should be incrementally distorted, we expect the generation to be also badly distorted. However, the observation do not match our expectation. We do not understand that the reviewer thinks why it can not be described as "failing to show the significance of exposure bias”. We designed this experiment to verify exposure bias, but the observations are not as expected, which is why we turn to more rigorous methods. We did not do any cherry-picking, and we did not say "this proves exposure bias is non-existent".
>
> - "However, it is unclear to me whether humans that are exposed to a generated prefix (which may already deviate somewhat from human language) would feel that the mistakes made due to exposure bias “match” the mistakes in the prefix. "
>
> A: In our human evaluation (Figure 3), we have explicitly asked turkers to judge the quality of the generation disregarding the context.
>
> - "the paper attempts to disprove the very existence of exposure bias... this overclaim, and the lack of smaller claims to build-up to it that might have been more appropriate,  ..."
>
> A: We did not say exposure bias is non-existent, and that's certainly not our "goal". To avoid this confusion, we have changed the text to make our conclusion more clear and structured (as suggested by the reviewer). Our main conclusion is two-fold: (1) We confirm that removing the training-generation discrepancy in the prefix does indeed improve the generation, but the gain is limited. (2) We do not observe the incremental distortion in the generation, which contradicts the claim of exposure bias. We believe this conclusion are reasonable given our consistent measurements.
>
> - "Why was wiki103 chosen instead of WebText used alongside the pretrained GPT-2?"
>
> A: wiki-103 has been a very popular large-scale LM benchmark for language modelling (https://arxiv.org/pdf/1901.02860.pdf). While we agree GPT-2 is also an important model to study, there's no obvious reason that the observations would be significantly different for GPT-2. In particular, since GPT-2 is trained by much larger amounts of data, we believe the impact from exposure bias can only be weaker for the GPT-2 model.

---

> > ### Comment · AnonReviewer2 · 2020-11-20
> > **further response**
> >
> > Thank you for the response.
> >
> >
> > > BLEU has been widely used for open-ended NLG (https://arxiv.org/pdf/1811.02549.pdf, https://arxiv.org/pdf/1609.05473.pdf). To our knowledge, there's no perfect existing automatic metric for NLG, which is why we we have repeated the EB-M experiments with a number of other metrics (Appendix B), and added the human evaluation. The measurements give consistent observations.
> >
> > This does not address my concerns about the way BLEU is used. The worry is not that BLEU is merely imperfect, but that it has (a) never been verified as truly measuring any background notion of quality when used in this way, much like this paper’s thesis about exposure bias never having been verified, and (b) is being used in a _fraction_, which means that there is a multiplicative interaction of any error from the use of BLEU.  I also agree with Reviewer 3’s concerns regarding _n_grams being insufficient to test exposure bias and that the human eval is unconvincing, in a similar way as I mention in my own review. In your response to Reviewer 3 you address why generations of length 30 are necessary to make evaluation feasible, but you do not address their concern that this will make D-prefix and M-prefix generations very similar to each other.
> >
> > > We provided the intuition of EB-C at the last paragraph of Sec4.1, where EB-C is proposed to cover the shortcoming (consistency aspect) of EB-M. Re "how big of a difference should we expect": We have repeated in the manuscript that assuming the incremental distortion from exposure bais, we expect the EB-C measurements to become bigger with longer prefix. However, in our experiment the measurements are stable around 1.01 or 1.02.
> >
> > In multiple responses you claim that the fact that exposure bias is incremental should make it more present the longer the sequence is. There is no reason this should be the case; there may be a ceiling to how bad strings get or it may fluctuate. Your argument appears to implicitly assume that if exposure bias is incremental it should be exponential over sequence length. Your experiments prove exposure bias is likely not exponential, but the claims you list in your response go significantly further than this claim.
> >
> > > The reviewer may have omitted our paragraph 2 of Section 3, where we give the intuition. And instead of "look at data", we look at generation from different kinds of prefixes. Here we repeat the intuition: We feed the model with (1) data prefix, (2) model prefix, (3) random prefix. In case (1), the training-generation discrepancy is removed from the prefix, so the generation should be much better than case (2). In case (3), since exposure bias claims that generation should be incrementally distorted, we expect the generation to be also badly distorted. However, the observation do not match our expectation. We do not understand that the reviewer thinks why it can not be described as "failing to show the significance of exposure bias”. We designed this experiment to verify exposure bias, but the observations are not as expected, which is why we turn to more rigorous methods. We did not do any cherry-picking, and we did not say "this proves exposure bias is non-existent".
> >
> > Thank you for pointing out paragraph 2 of Section 3. I did, indeed, read it, and quote the entirety of it here:
> >
> > > The intuition behind the prefix-switching experiment follows immediately from the original claim of exposure bias: During generation, if we set the prefix distribution to be the ground-truth data distribution instead of the model’s own distribution, then the discrepancy between training and generation in the prefix is removed, and hence the model’s generation quality should be much better. In the extreme case of shuffled or random prefixes, due to the claim from exposure bias that the errors should accumulate, we expect the model to generate also badly distorted sequences.
> >
> > This is a good intuition of what should happen, but it is not a rigorously testable hypothesis. Perhaps the most important two variables left unspecified is how errors should “accumulate” (exponentially? linearly? like a sigmoid?) and what does “badly distorted sequences” mean? I fully believe that the authors could tell if the generations were complete gibberish, but even in the absolute worst case one would expect models to back-off into reasonable trigrams, which could read well enough. Furthermore, you write:

---

> > > ### Comment · AnonReviewer2 · 2020-11-20
> > > **further response (continued)**
> > >
> > > > wiki-103 has been a very popular large-scale LM benchmark for language modelling (https://arxiv.org/pdf/1901.02860.pdf). While we agree GPT-2 is also an important model to study, there's no obvious reason that the observations would be significantly different for GPT-2. In particular, since GPT-2 is trained by much larger amounts of data, we believe the impact from exposure bias can only be weaker for the GPT-2 model.
> > >
> > > This seems to imply that even D-prefix generations should not be as high-quality as currently available models, so perhaps the gap between “badly distorted sequences” and baseline generations is simply not that large?
> > >
> > > > In our human evaluation (Figure 3), we have explicitly asked turkers to judge the quality of the generation disregarding the context.
> > >
> > > While I appreciate that this was part of the given instructions, humans are extremely sensitive to context. Were they shown the D- or M-prefixes? Simply asking annotators to ignore given context does not remove human biases.
> > >
> > > > We did not say exposure bias is non-existent, and that's certainly not our "goal". To avoid this confusion, we have changed the text to make our conclusion more clear and structured (as suggested by the reviewer). Our main conclusion is two-fold: (1) We confirm that removing the training-generation discrepancy in the prefix does indeed improve the generation, but the gain is limited. (2) We do not observe the incremental distortion in the generation, which contradicts the claim of exposure bias. We believe this conclusion are reasonable given our consistent measurements.
> > >
> > > In regard to (1) please see the previous concern about metrics that I have elaborated in my response. As for (2), I do not believe that the notion of “incremental distortion” has been properly defined in this paper. To disprove the presence of “incremental distortion” it must be properly formalized, and the question of how much distortion should increase and what this should look like has not been convincingly argued here, despite some relevant intuitions being present in the paper.

---

### Official Review · AnonReviewer4 · 2020-10-27
**Review for: Quantifying Exposure Bias for Open-ended Language Generation**

**Rating:** 6
**Confidence:** 4

**Review:**

Summary:

The so-called "exposure bias problem" (EBP) is often cited as a serious issue when training sequential model with MLE and teacher forcing. This paper attempts to quantify whether the problem is real on open-ended generation experiments, and concludes that it is actually minor.

Positive aspects of the paper:

- EBP is often accepted as an obvious problem in the generation community, despite the lack of experimental evidence. Thus, a paper such as this one that tackles such evidence head-on can be very useful.

- Close to the end, the paper argues, I think correctly, that "the original claim of exposure bias is not well defined, [and] our approaches can only act as a reasonable proxy to measure its seriousness". The paper however attempts to design (1) experiments and (2) quantitative measures that correspond with our general intuition of the EBP.

- According to these experiments and measures, and also to some human evaluations, the paper compares two situations: (D) the trained generator is provided with a prefix from (an unseen portion of) the dataset and asked to generate a continuation; (M) it is provided with a prefix that it generated itself and asked to generate a continuation. The conclusion from these experiments is that the continuations in the (M) case are only very slightly worse than in the (D) case. This contradicts the usual expectation about the EBP, namely that in the (M) case, the model would be "lost" in unknown territory and start to produce very poor text. The main conclusion of the authors is that: "although the mismatch between the data and model prefix distribution exists, it is still in the model’s “comfortable zone”, and is not large enough induce drastic performance loss during generation".

- The authors are careful to avoid a possible misunderstanding of their results. They do *not* make the claim that MLE + teacher forcing is the best way to train a generation model, but only the *different* claim that exposure bias is not such a serious problem as is often assumed.

Some issues and questions:

- The fact that the generation model does not move away from its "comfort zone" when generating prefixes could be related to two different dimensions: First, the prefixes that you generate are only moderately long, thus alleviating the EBP issue. Second, you do not describe exactly how the generation is done. I gather that Transformer-XL uses a form of "top-k" sampling, that is, a generation mode more restricted than a standard (from the pure probabilistic viewpoint) "ancestral sampling". Such restricted sampling (as also beam-search) is known to improve the quality but limit the diversity of the generations. It would be interesting to see if you would obtain the same EBP conclusions with ancestral sampling.

- The corpus-BLEU measure that you use is most of your experiments was designed for Machine Translation and appears to me to be a very weak measure of quality for open-ended generation. It would actually be informative to compute a baseline for that measure in terms of continuations not from the model, but from the training data itself, giving a measure of the average "quality" of a gold-standard continuation relative to all the other gold-standard continutations from the training data. How good would this "gold corpus-BLEU" be? Probably not so good, and it would be interesting to compare the (D) and (M) continuations to these gold continuations.


Minor points:

- The JS divergence is not a distance.

- Please clarify exactly when "prompts" are used in your experiments. While they are used in the human evaluations, it was not clear to me whether they were used in the EB-M experiments, for example.

- The paper https://arxiv.org/abs/1906.05664 "Calibration, Entropy Rates, and Memory in Language Models" might be relevant as related work: in that paper the authors argue that neural language models tend to suffer from "entropy drift", namely the tendency to entropy of the next token prediction to be higher when conditioned on an (M) type prefix than on a (D) prefix. However, if I am correct, they assume standard hierarchical sampling, and their empirical evaluations are not extensive.

-----

Nov. 30th: On a second reading of the authors’ exchanges with the reviewers as well as of the updated paper, I am lowering my overall score. While I still believe that the *questions* that the paper raise are very worthwhile to the community, I agree with several reviewers that the *answers* provided in the paper are insufficiently supported by a convincing formalization and by experiments.

---

> ### Author Response · Authors · 2020-11-14
> **thanks for the review**
>
> Thanks for the review. We first note our changes in the manuscript: We have expanded the beginning of Sec4 and the final conclusion part, for a more structured interpretation of our methodology / results. There are also added citations, discussions (Sec7) and notation changes which are suggested by the reviewers. Here's our replies to the concerns:
>
> - "Such restricted sampling (as also beam-search) is known to improve the quality but limit the diversity of the generations. It would be interesting to see if you would obtain the same EBP conclusions with ancestral sampling."
>
> A: Sorry about the confusion, but we actually do use the ancestral sampling as you suggested. The reasons are as follows: (1) These sampling algorithms are known to trade quality out of diversity. So, invoking them could "hide" the exposure bias problem because the prefixes from the model will be of higher quality. (2) The sampling algorithms requires tuning of hyper-parameters, which will complicate the comparison (for different prefix length).
>
> - "corpus-BLEU could be weak"
>
> A: BLEU has been widely used for evaluating open-ended NLG models (https://arxiv.org/pdf/1811.02549.pdf, https://arxiv.org/pdf/1609.05473.pdf). To our knowledge, there's no perfect automatic metric for NLG, which is why we added the human evaluation. We agree BLEU has weakness, but we have repeated the EB-M experiments with a number of other metrics (Appendix B), and the measurements are consistent.
>
> - "The JS divergence is not a distance."
>
> A: Yes, you are right and thanks for pointing this out. We have fixed it in the manuscript.
>
> - "Please clarify exactly when "prompts" are used in your experiments."
>
> A: The prompts are used in human evaluation to restrict the topics, and reduce the variance in the comparison. They are not used in the EB-M or EB-C experiments.

---

> > ### Comment · AnonReviewer4 · 2020-11-22
> > **Thanks to the authors for clarifying certain points and for updating the paper.**
> >
> > Thanks to the authors for their responses to the different reviews and for updating their paper.

---

### Official Review · AnonReviewer1 · 2020-10-29
**Solid empirical study of an important question**

**Rating:** 6
**Confidence:** 4

**Review:**

This paper presents an empirical study of exposure bias, showing that it does not appear to be an especially significant issue.  Both automated metrics and human evaluations are presented, and I found the experiments to be fairly convincing.  While the paper could have more material and depth, I think it makes an important point that people will appreciate seeing in the conference.

Designing metrics for exposure bias is a novel task, and this paper invented appropriate approaches.  The experiments cover the two most important model classes (LSTMs and transformers) and use representative model settings and corpora.

Concerns:
The experiments only look at pure sampling from the model---i.e., they don't use temperature, nucleus sampling, greedy or beam search.  These other generation settings are more important for applications, compared to pure generation.  Also, they have been reported to exhibit more deviation from the corpus distribution (especially with respect to particular pathologies like repetition).  Evaluating generation in these other settings would increase the impact of the paper's conclusions.

The human evaluation is helpful, but needs measures of inter-annotator agreement.

Less significant: The qualitative experiment that starts the paper is very anecdotal, and I think the paper would be better off without it.  The concrete examples in Table 1 are helpful for illustration, but I would not dedicate a section to them or call it an ‘experiment,’ as it is too limited in scope.  It detracts from the stronger experiments later in the paper.  Likewise, the model in Example 1 seems too inaccurate to be illustrative, and the associated discussion in Appendix D is already well known I believe (that the “Teacher forcing” MLE objective just aims to choose parameters that maximize the likelihood of the corpus, using the chain rule).

---

> ### Author Response · Authors · 2020-11-14
> **thanks for the review**
>
> Thanks for the review. We first note our changes in the manuscript: We have expanded the beginning of Sec4 and the final conclusion part, for a more structured interpretation of our methodology / results. There are also added citations, discussions (Sec7) and notation changes which are suggested by the reviewers. Here's our replies to the concerns:
>
> -"The experiments only look at pure sampling from the model---i.e., they don't use temperature, nucleus sampling, greedy or beam search."
>
> A: we did not use temperature or nucleus sampling for the following reasons: (1) These sampling algorithms are known to trade quality out of diversity. So, invoking them could "hide" the exposure bias problem because the prefixes from the model will be of higher quality. (2) The sampling algorithms requires tuning of hyper-parameters, which will complicate the comparison (for different prefix length). (3) The repetition problem mentioned by the reviewer happens for both common (e.g., "I don't know") and uncommon prefixes (https://arxiv.org/abs/1904.09751). Therefore, we believe the link to training-generation discrepancy is minimal. But we agree it's an important issue for NLG models.
>
> -"The human evaluation is helpful, but needs measures of inter-annotator agreement."
>
> A: As requested by the reviewer, we computed that the inter-annotator agreement (Cohen's kappa ) for annotations in our human evaluation is around 70% for different configurations. We added this to the manuscirpt, thanks.

---

> > ### Comment · AnonReviewer1 · 2020-11-24
> > **Thanks for the response**
> >
> > I do think that looking at other generation approaches would improve this paper.  Also, to address the other reviewers' concerns about the quality of the (relatively untested) metrics used here, perhaps some experiments that calibrate the measures would be helpful.  For example, how much better is medium LSTM on these EB measures, compared to a small LSTM?  If different models exhibit much larger differences on these metrics compared to what you see when swapping in a real prefix with a generated one, in my opinion that's a convincing argument that exposure bias is relatively unimportant, if it has less impact on metrics than just scaling up the models somewhat.

---

> > > ### Author Response · Authors · 2020-11-25
> > > **Thanks for the response**
> > >
> > > Hi, we want to mention that Table 2 (the EB-M results) contains the sort of comparison you suggested. By comparing the BLEU scores, we get that
> > >
> > > (1) Switching to shuffled prefix gives much worse BLEU scores, for the transformer model.
> > >
> > > (2) The BLEU score from the LSTM model is much worse than the transformer model (this is in spirit, similar to your proposed comparison of a small LSTM to a medium LSTM).
> > >
> > > Finally, the gap induced by switching prefix from model to data (removing the discrepancy in the prefix) for LSTM or transformer, is much smaller than the gap induced by the listed two switching. Therefore we conclude that this set of experiments suggest that the impact from exposure bias is weak or limited. Thanks again for the response & Suggestion!

---

### Official Review · AnonReviewer3 · 2020-11-01
**Not convinced by the evaluation**

**Rating:** 3
**Confidence:** 4

**Review:**

The paper studies the exposure bias in auto-regressive neural language models. This problem is known to cause incremental performance degradation, and attempts to mitigate this problem have received significant attention in the community (using, e.g., RL and GANs). The paper claims that prior work has mostly focused on addressing the problem rather than measuring how severe the exposure bias problem actually is. Despite extensive previous work on mitigating exposure bias, the paper suggests that the exposure bias is not “large enough [to] induce drastic performance loss during generation” (e.g., a human evaluation controlling for exposure bias show relative differences of only < 3%).

On the positive side, I agree with the paper that it is worthwhile to study the extent of the exposure bias problem, and the approach of the paper (comparing generation with model-based prefixes vs. data prefixes) is quite intriguing, as it attempts to compare generation with exposure bias again without such a bias. That said, I have several concerns that make me question some of the claims of the paper:

1) The human evaluation shows very little performance difference between generation with data prefix (D-prefix) and model prefix (M-prefix), suggesting exposure bias is not a problem. But I would argue this is mostly an artifact of the experimental setup, as prefixing generation with D-string (of length L) only eliminates exposure bias up to position L, and then both evaluated systems (according to the generation process defined Section 4.1) use a standard auto-regressive LM generation process that makes them both subject to the exposure bias problem. Since these generated strings are of length 30, there is plenty of room for exposure bias to crop in. (Indeed, Zhang et al., (2019) and Holtzman et al. (2019) have shown concrete examples of exposure bias artifacts appearing in much shorter sequences). So, the small performance difference between a preference M-prefixed and D-prefixed generation may very well be due to both setups being almost identical (same model, same auto-regressive inference algorithm, and the only difference is the prefix – which human raters are not even asked to judge directly). If exposure bias is indeed a problem, then it would affect both systems almost the same way, so this human evaluation can’t be used to either affirm or deny exposure bias is a significant problem.

2) I also have concerns regarding the automatic evaluation based on BLEU. What the authors call “corpus-BLEU” is actually not the standard version of BLEU (see “Other Comments” below), as the version of the paper looks at n-gram (n=1 to 3) matches between the generated sentences and a large set of references that is *not* specific to a particular context. As the paper’s automatic evaluation is done in a completely context-agnostic way and relative to a large pool of references, it essentially only measures whether the model (with or w/o exposure bias) is able to generate plausible trigrams, but that sets the bar very low as we already have plenty of evidence showing neural language models are quite capable of generating reasonable trigrams (whether there is exposure bias or not), and in fact often much longer n-grams. The authors claim that auto-regressive models appear to have “self-recovery ability” that mitigate any exposure bias, but I would say that it is hard to claim anything about self-recovery (at least in terms of automatic evaluation) when the evaluation metric operates over such a small window (<=3 words).

3) Abstract: “Although a lot of algorithms have been proposed to avoid teacher forcing and therefore alleviate exposure bias, there is little work showing how serious the exposure bias problem actually is.”
I find this claim a bit misguided, as the numerous papers addressing exposure bias are *empirical* ones. They have shown improvements in various tasks such as machine translation, image captioning, and other generation tasks thanks to techniques aimed at reducing exposure bias. Now, one could claim that significant improvements shown in these papers are due to reasons other than exposure bias, but if that is the case then the submission doesn’t do a much better job at isolating exposure bias from other factors (given my concerns in (1) and (2)). Note that Zhang et al. (2019) actually does include a short section attempting to quantify the effect of exposure bias.

4) For a paper that attempts to challenge the current understanding of a problem that has received very significant attention (i.e., exposure bias), it is quite thin in terms of related work (2 paragraphs). The paper omitted important related work (e.g., Schmidt, 2019; Tan et al., 2019; Rennie et al., 2016), including an ACL best paper on the same topic and whose findings appear to be at odds with the current submission.

In sum, the paper tries to improve our understanding of exposure bias and its impact on open-ended language generation, and this is totally a worthy goal. I also recognize that isolating (i.e., ablating) the effect of exposure bias is difficult. That said, the paper makes rather strong claims (e.g., that “performance gain is minimal” if exposure bias is supposedly eliminated) that I find unfounded given my points in (1) and (2). In both evaluations, the setups evaluate a given model sequence W prefixed by a string sampled either from the data or the model, but that does not eliminate the fact that exposure bias is bound to appear within the generated sequence (length 30), which is the sequence that is evaluated in the end.

Other comments:

“Corpus BLEU”: Note that BLEU, as originally defined in (Papineni et al., 2002), is in fact a corpus-level metric, as it aggregates n-gram statistics over an entire (test) corpus. So “corpus” in “corpus BLEU” is redundant and possibly confusing. The term “corpus BLEU” or “corpus-level BLEU” is generally used to contrast with various versions of “sentence-level BLEU.” Now it appears that “corpus BLEU” in this submission refers to the version of BLEU used in SeqGAN (Yu et al.), which is therein not called “corpus BLEU.” That distinction should be made clearer, considering that the use of “a large number of sentences from ground-truth data as references” is a significant departure from how BLEU was originally designed to work. Indeed, (corpus-level) BLEU (Papineni et al., 2002) doesn’t allow matching hypotheses and references across test instances, as opposed to the submission. The authors justify their use of BLEU as it is a “well-established [metric] in the NLG literature,” but this is rather misleading as their specific version of BLEU is not well established and not what is commonly used in MT and NLG.

Table 1 doesn’t actually illustrate what the experimental setup of the paper does (i.e., Section 4.1), as the latter doesn’t include a shared prompt of 20 words that is supposed to make the two generated strings “more comparable”. Since this prompt supposed to make the two strings more comparable is absent from the actual evaluation setup, I gather from the authors’ own words that they implicitly admit that string comparisons in their evaluation setup are not so comparable.

* Zhang et al., 2019: https://arxiv.org/pdf/1906.02448
* Schmidt, 2019: https://arxiv.org/abs/1910.00292
* Rennie et al., 2016: https://arxiv.org/abs/1612.00563
* Tan et al., 2019: https://arxiv.org/abs/1811.09740
* Holtzman et al., 2019: https://arxiv.org/abs/1904.09751
* Chen and Cherry, 2014: https://www.aclweb.org/anthology/W14-3346/

---

> ### Author Response · Authors · 2020-11-14
> **Thanks for the review.**
>
> Thanks for the review. We first note our changes in the manuscript: We have expanded the beginning of Sec4 and the final conclusion part, for a more structured interpretation of our methodology / results. There are also added citations, discussions (Sec7) and notation changes which are suggested by the reviewers. Here's our replies to the concerns:
>
> - "Since these generated strings are of length 30, there is plenty of room for exposure bias to crop in...it would affect both systems almost the same way"
>
> A: We choose the generation to be length 30, so that the human evaluation can be effective. If we use a length, say, 10, then in the generation there usually won't even exist a complete sentence, and it will be hard for annotators to rate. We agree that in the generation, EB could "crop in" for the data prefix case. But again, exposure bias claims that the generation should be **incrementally** distorted, and we use a long prefix up to length 60. So the **degree** of distortion from EB, should be much more serious in the model prefix case than data prefix, even when EB has "cropped in" for both cases.
>
> - "Indeed, Zhang et al., (2019) and Holtzman et al. (2019) have shown concrete examples of exposure bias artifacts appearing in much shorter sequences"
>
> A:  The repetition problem (Holtzman et al. (2019)) happens for both common (e.g., "I don't know") and uncommon sequences (https://arxiv.org/abs/1904.09751). Also, the training-generation discrepancy is not discussed in that paper. Therefore, we believe the link to exposure bias is minimal. But we agree it's an important issue for NLG models. Re Zhang et al., (2019), please see the reply below.
>
> - "Note that Zhang et al. (2019) (ACL best paper) actually does include a short section attempting to quantify the effect of exposure bias."
>
> A: We suppose the reviewer is referring to Sec5.7 of "https://arxiv.org/pdf/1906.02448.pdf", where they attempt to measure the gain of solving exposure bias by "count the ground truth words whose probabilities in the predicted distributions produced by our model are greater than those produced by the baseline model, on 1k training data". However, the intuition of how this counting can confirm that their gain is from alleviating exposure bias is unclear. If anything, this only suggests that their proposed model performs better than the baseline model on the training data (prefixes). We have added this discussion to the manuscript.
>
> - "BLEU has weakness (is context-agnostic)"
>
> A: BLEU is a widely used automatic metric in NLG literature. We agree BLEU has weakness, but we have repeated the EB-M experiments with a number of other metrics (Appendix B), and the measurements are consistent. To our knowledge, there's no perfect automatic metric for NLG, which is why we added the human evaluation. And our proposed EB-C captures the consistency aspect, which is not context-agnostic.
>
> - Notation of corpus-BLEU
>
> A: We agree our notation could be confusing, we have changed it to BLEU (following https://arxiv.org/pdf/1609.05473.pdf). And by "well-established", we mean "popular". We have changed it in the manuscript, thanks for pointing this out.
>
> - The author omitted Schmidt, 2019; Tan et al., 2019; Rennie et al., 2016;
>
> A: Thank you for the reference. We have added citations and discussion of these works to the manuscript, yet most of them  (similar to other non-MLE works) only assume the seriousness of EB without verifying it (We have pointed out this in the Intro Section).
>
> - Table 1 doesn’t actually illustrate what the experimental setup of the paper does (i.e., Section 4.1), as the latter doesn’t include a shared prompt of 20 words that is supposed to make the two generated strings “more comparable”.
>
> A: We did not say Table 1 is for illustrating Sec4.1. Table 1 is a qualitative comparison (Sec3) and we wish to make it easier for readers to compare the samples. That's why we let the model prefix and data prefix share the same prompt so that the generation will likely be on the same topic. On the contrary, the EB-M experiments (Sec4.1) are quantitative. We are not comparing generations side-by-side but rather quantify their quality/diversity by various metrics, thus there's no need for the shared prompts.

---

### Decision · Program_Chairs · 2021-01-07
**Final Decision**

**Decision:**

Reject

**Comment:**

Sequence generation models trained via maximum likelihood estimation (or variants of so called 'teacher-forcing') condition on *data* samples during training and on *model* samples for predictions. The susceptibility to this potential "mismatch" in input distribution is often referred to as exposure bias (EB).

This paper stresses that most research around EB is focused on addressing it, rather than defining and/or quantifying it. Thus the submission questions the severity of EB and attempts to operationalise a testable definition for it. Myself and all the reviewers strongly support the observations and the agenda, we find the question this paper asks an important one.

Despite our appreciation for this paper's relevance, we have identified a number of problems that prevent me from recommending this paper. I will comment on the two most important points:

1. The 'operational definition' of EB in this paper is not sufficiently precise to be testable. It builds on the somewhat commonly accepted view that the effects of EB accumulate as the conditioning context grows longer, and that this causes a model to generate badly distorted sentences. This definition still leaves quite some room for interpretation (without specifying reasonable expectation about how these effects 'accumulate' and what/how bad they are, it seems difficult to design tests). We acknowledge that the submission attempts to shed light onto some of these aspects by having some 'control groups' using gold data and shuffled strings, but we did not find those sufficient (mostly in light of the next point).

2. MT evaluation metrics (essentially, string similarity metrics), most notably (but not exclusively) BLEU, are used in this work in a setting where we cannot easily grant that they have the discriminating power that the authors expect of them. See this is not a criticism about the imperfections of BLEU (or any other automatic metric), but about the lack of evidence supporting its use against unrelated sentences. We do not find it sufficient that some recent NLG papers have made similar use of it (I, for example, would have criticised those papers on similar grounds).

Overall, we believe this submission asks a relevant question, the insight about dependence on prefix is nice and might lead to a first operational definition of EB (which might be only a few refinements away from the version proposed here). The current evaluation is unconvincing and I believe the authors should be able to find more credible strategies, especially, strategies that have already gone through some scrutiny (for example, in literature around OOD detection and tests for distribution shift).

Though I do not recommend this paper for acceptance, I hope the authors will find valuable feedback in the expert reviews attached.